# A Comparison of the Immediate Effects of Verbal and Virtual Reality Feedback on Gait in Children with Cerebral Palsy

**DOI:** 10.3390/children11050524

**Published:** 2024-04-27

**Authors:** Tine De Mulder, Heleen Adams, Tijl Dewit, Guy Molenaers, Anja Van Campenhout, Kaat Desloovere

**Affiliations:** Clinical Motion Analysis Laboratory, KU Leuven, 3000 Leuven, Belgium; heleen.1.adams@uzleuven.be (H.A.); tijl.dewit@uzleuven.be (T.D.); guy.molenaers@uzleuven.be (G.M.); anja.vancampenhout@uzleuven.be (A.V.C.); kaat.desloovere@uzleuven.be (K.D.)

**Keywords:** gait analysis, rehabilitation, virtual reality, verbal feedback, cerebral palsy

## Abstract

Different types of feedback are used during gait training in children with cerebral palsy (CP), including verbal (VB) and virtual reality (VR) feedback. Previous studies on VR feedback showed positive effects on the targeted gait parameter. However, both positive and negative side effects on other parameters were seen as well. The literature on the effect of VB feedback is lacking and, to our knowledge, both feedback methods have not yet been compared. In this monocentric study with a single-session intervention protocol, children with CP completed a training session on the Gait Real-Time Analysis Interactive Lab (GRAIL) and received both VB and VR feedback on hip extension, in randomized order. Outcome parameters were continuous gait curves of sagittal kinematics and hip kinetics, specific features of hip angle and moment, sagittal gait variable scores and gait profile scores. Improvement of the targeted gait parameter was seen both after VB and VR feedback, with a small advantage for VR over VB feedback. Furthermore, positive side effects on knee and ankle sagittal kinematics were seen. However, the overall gait profile score did not improve, most likely due to negative compensatory strategies. In conclusion, children with CP can adapt gait in response to both VB and VR feedback, with VR feedback producing a slightly better effect. Due to secondary effects on parameters other than the targeted parameter, the overall gait did not improve.

## 1. Introduction

Achieving and improving independent gait is the main goal of treatment in children with Cerebral Palsy (CP) as it improves body function and participation in activities of daily life. Treadmill training has been proven to be a clinically effective intervention to enhance gait function in children with CP, as it allows for repetitive, task-specific exercises with the ability to control walking speed [1,2]. Compared to overground walking exercises, treadmill training causes larger improvements in functional mobility, performance and balance [3]. Furthermore, enhanced range of motions of the ankle and hip with decreased pelvic rotation were seen in children with CP when walking on a treadmill compared to overground walking [4].

During training, different types of feedback, such as verbal (VB) and virtual reality (VR) feedback, are used to improve specific gait deviations in children with CP. The literature on the effect of VB feedback on gait in patients with CP is limited. Spatiotemporal and kinematic parameters were found to be improved after auditory stimulation in subjects with CP. These auditory signals included music following the speed and rhythm of the participant or a clicking sound at heel strike, amongst others [5,6,7]. On the contrary, instructional quotes combined with descriptive pictures on different movement patterns needed to complete a specific motor task generated very little effect on gait in children with CP [8]. Other research on VB feedback has mainly focused on patients recovering from a stroke or patients with Parkinson’s disease, with positive effects on muscle activity, step length and velocity. However, other studies reported negative outcomes, such as an increased knee flexion during mid-stance, or no effect at all [9,10,11]. We can conclude that, thus far, results on the effects of VB feedback have been inconsistent. To our knowledge, no previous study has investigated the effects of VB feedback on gait in children with CP during training on a treadmill.

The literature regarding the effect of VR feedback on gait in children with CP is more extensive. Positive effects are described both during single treadmill training sessions as well as after a training program of four or eight weeks. Positive effects include increased muscle strength, balance and gross motor function. Furthermore, improved spatiotemporal parameters and kinematics were described, including increased hip and knee extension in stance and peak ankle power generation, among others [12,13]. Interestingly, feedback on one gait parameter also showed indirect impact on other parameters. Moreover, although several gait parameters improved, there was no change in overall gait performance, as measured by the Gait Profile Score (GPS) [14,15]. In conclusion, positive effects on different gait parameters have been described, both after single-session and long-term VR training programs in children with CP. This demonstrates the ability of these children to adapt their walking pattern while receiving VR feedback. However, feedback on one gait parameter influences other parameters, either positively or negatively, and an evident improvement in overall gait performance has not yet been documented.

VR feedback is expected to have some theoretical advantages compared to VB feedback. Motor rehabilitation requires multiple task repetitions which can be monotonous and can lead to non-compliance. Virtual reality provides an enjoyable and motivating environment in which the user is constantly and progressively challenged although performing the same underlying task [16]. However, VR feedback is more expensive and requires more extensive infrastructure, whereas VB feedback is less costly and always at hand. To our knowledge, the effects of both feedback methods on gait in children with CP have not yet been compared.

During gait training, the choice of optimal feedback parameter is important, as children with CP do not experience every parameter as achievable and intuitive [14]. Feedback on hip extension has previously been described as comprehensive and effective [15]. Furthermore, excessive hip flexion is a common and energy-consuming gait deviation in children with CP [17,18]. Additionally, it is expected that an improved hip extension leads to a greater step length.

Therefore, the aim of the current study was to evaluate and compare the immediate effects of VB and VR feedback on hip extension during treadmill training in children with CP. We first hypothesized that walking on a treadmill with VB or VR feedback in children with CP results in improved gait (i.e., closer to the gait pattern of TD children) when compared to walking on a treadmill without feedback. Secondly, it was hypothesized that VR feedback leads to better gait outcomes than VB feedback.

## 2. Materials and Methods

### 2.1. Subjects

All children who were scheduled for a routine clinical gait analysis in the Clinical Movement Analysis Laboratory (CMAL) of the University Hospital of Leuven, Belgium, between October 2018 and December 2021 and who fulfilled all inclusion criteria were invited to participate in the current study. Children were selected for the study if they met the following inclusion criteria: predominantly spastic type of CP, uni- or bilateral involvement, GMFCS level I or II (walking without an assistive device) and sufficient cognitive skills allowing them to carry out instructions. Patients were excluded if they had received surgery on the lower limbs, selective dorsal rhizotomy or baclofen treatment less than 12 months prior to the assessment; received Botulinum neurotoxin type A (BoNT-A) less than six months prior to the assessment; had a visual or hearing deficit that could interfere with the ability to interpret VB or VR feedback; or had behavioral problems that might negatively influence the cooperation. Participation in the study was voluntary. All participants and/or their parents signed an informed consent. The study was conducted in accordance with the Declaration of Helsinki and received approval of the Medical Ethical Committee of the University Hospital of Leuven. All included patients were assigned to a Patient Study Code at inclusion to ensure privacy of the participants to all researchers except the second author, who was responsible for data collection.

The sample size calculation was based on the data of a preliminary pilot study that was performed on 5 children with CP who received VR feedback on hip extension. In this pilot study, the peak hip extension angle averaged 5.77° (±4.47) at baseline and 3.11° (±3.62) after VR feedback. The effect size measured 0.78 (calculated following the Cohen’s d test) for peak hip extension angles in the virtual reality feedback condition compared to baseline condition [19]. G*Power software (https://www.psychologie.hhu.de/arbeitsgruppen/allgemeine-psychologie-und-arbeitspsychologie/gpower 2022–2023) was used to calculate sample size, based on a two-sided, one-sample dependent *t*-test for differences in means, with power (1-β) = 0.90 [20]. To take multiple tests into account, the significant level α was adjusted to 0.01. The minimum number of subjects required to detect statistically significant differences between baseline and feedback conditions for the total group was 28.

### 2.2. Study Design

Before the feedback intervention protocol, all children received a routine overground clinical gait analysis, which was used to define the patient-specific preferred walking speed, and which included a standardized clinical examination. The preferred overground walking speed was used to define the initial speed during the habituation period, which was set at a similar pace to the overground walking speed, i.e., 1.07 m/s (±0.20) on average. The walking speed was then further modified until the children felt comfortable while walking on the treadmill, which was then held constant during the remainder of the intervention protocol. During the clinical examination, joint range of motion, spasticity and strength at the lower limb joints (hip, knee and ankle) were assessed. Spasticity was assessed using the Modified Ashworth Scale (MAS) [21] and Tardieu scale [22], while strength of the flexors and extensors at the hip, knee, ankle and hip abductors and adductors was classified according to the Medical Research Council Muscle Strength Scale [23].

Afterwards, all patients performed a single-session intervention protocol on the Gait Real-Time Analysis Interactive Lab (GRAIL) (Motek Medical BV, Amsterdam, The Netherlands). The GRAIL combines a force-sensing split-belt treadmill with a virtual reality gaming environment and a 3D motion capture system (Vicon, Oxford, UK). The Human Body Model (HBM) and D-Flow software (Motek Medical BV, Amsterdam, The Netherlands https://www.motekmedical.com/software/d-flow/ 2022–2023) were used to calculate real-time gait parameters and to obtain real-time VR feedback. The Plug-In-Gait model and Vicon Nexus software (Vicon, Oxford, UK https://www.vicon.com/software/nexus/ 2022–2023) were used to acquire kinematic and kinetic data and to perform data processing.

An overview of the test protocol is given in Figure 1. The protocol started with a four-minute habituation period to allow the participant to adapt to treadmill walking and to determine a comfortable walking speed. When the patient was feeling comfortable on the treadmill, a baseline trial of one minute was acquired, which allowed to determine the maximal hip extension. The baseline measurement was used to choose the most affected leg for the feedback trials and to define the target hip angle during VR feedback.

Following the baseline measurement, the patient received two feedback sessions. The patients received feedback on hip extension at late stance phase and on ankle power generation during push off. The order of the targeted gait parameters was randomized between patients, as determined by a coin toss. Feedback on ankle power generation was found to be too challenging to comprehend and to properly respond to for a significant number of the enrolled subjects. More specifically, the timing of response to this type of feedback (within the gait cycle) was found to be the most difficult factor. Moreover, VR feedback on ankle power yielded variable results both between and within participants. Feedback on this parameter has indeed been proven to be experienced as more difficult and less intuitive when compared to other feedback parameters, such as knee extension and step length [14]. Further research is necessary to determine how this type of feedback can be improved, and which children can benefit from feedback on ankle power, before this can be adequately implemented as feedback intervention. Therefore, the current study only analyzed the results regarding hip extension. In each feedback session, the patient received three minutes of VR feedback, three minutes of VB feedback (in a random order) on their most affected leg and walked one minute without feedback. The order of VR and VB feedback was switched in the second feedback session.

All children received VB feedback from one of three researchers at the gait analysis laboratory. This person was standing next to the treadmill with a sagittal view of the participant. Standardized VB feedback instructions were given based on visual inspection of the gait. These instructions were either affirmative or instructional and were given at each consecutive step. No tactile feedback was added when VB feedback was given.

During VR feedback, an avatar portrayed the subjects’ real-time movements and provided visual and auditory rewards as motivation to achieve the targeted gait modification. An example of such VR feedback is pictured in Figure 2.

The initial target was set at an increase of 5° hip extension. If the task to reach the target was found to be too easy (i.e., when the target was reached ten successive steps within the first fifteen steps) or too difficult (i.e., when the target was not reached within the first fifteen steps), the target was manually adjusted to maintain motivation and to ensure maximum improvements. The final target for hip extension during VR feedback was the minimal hip extension angle during baseline walking minus 5 degrees (to encourage the children to exceed their habitual hip extension deficit, but still ensure feasibility of a successful score). All children walked barefoot.

### 2.3. Data Collection and Data Analysis

Further data analysis was only performed on the targeted leg. Kinematic and kinetic datasets were obtained via Vicon Nexus software (Oxford, UK). All segmented datasets were time-normalized to the gait cycle duration. The amplitude of power and moments were also normalized to body weight. Joint angles and moments were decomposed in the three anatomical planes (sagittal, coronal and transversal). Since only feedback on sagittal hip motion was evaluated, mainly sagittal data were analyzed for the current study. Yet, the Gait Profile Score (GPS) quantified the combined feedback impact, i.e., on all motion planes. The primary focus was on the effect of feedback on sagittal kinematics, whereas, in regard to kinetics, only the effect on hip moment was analyzed. All processed data were imported into the custom-made multiple joint software MATLAB R2019 (The MathWorks, Natick, MA, USA) to obtain continuous waveforms. Each waveform was interpolated to intervals of 2%, which provided a total of 51 data points for each gait cycle waveform. Ten gait cycles per minute were defined for further data processing, i.e., ten gait cycles for the baseline condition and 30 gait cycles for each of the feedback conditions. Equal distribution of the steps over the three minutes of the feedback conditions was ensured to rule out potential bias for learning- or mental-fatigue effects. Further data processing consisted of two quality checks. A schematic overview is shown in Table 1. We first evaluated the quality of the gait cycles through visual inspection, and obvious outliers and artefacts were excluded. This resulted in an unequal number of outliers per condition and participant. We then aimed for an equal number of steps per minute for all participants, which was seven gait cycles for the baseline trial and for each of the three minutes of the feedback trials. In case more than seven good-quality steps were available per subject and per minute, we excluded the steps that deviated the most from the averaged waveform. As a result of this data exclusion, the final data analysis was performed on seven steps for the baseline and 21 steps for the feedback condition (seven steps per minute).

Three sets of outcome parameters were defined for data analysis: (1) five clinically relevant continuous gait curves (four curves on sagittal kinematics: sagittal ankle, knee, hip and pelvis angles; and one kinetic curve of internal hip moment), (2) six discrete gait features (i.e., the minimal hip angle, the timing within the gait cycle where the internal hip extension moment transfers to hip flexion moment (hip moment 0) and the Gait Variable Scores (GVS) for the ankle, knee, hip and pelvis) and (3) the Gait Profile Score (GPS) as measures of overall gait performance [24].

For further statistical analysis, a comparison was made between the averaged outcome parameters of all included steps of the baseline condition and all included steps of each feedback condition. Statistical analyses of the continuous gait curves were performed with Statistical Parametric Mapping (SPM; SPM1d, version 0.4.3) in MATLAB 2018 (The MathWorks, Natick, MA, USA) as it allows hypothesis testing over entire waveforms. Critical thresholds were calculated using random field theory. When these thresholds were crossed, suprathreshold clusters were established which define the regions within the gait cycle where statistical evidence was identified. For each suprathreshold cluster, the extent of the cluster and the *p*-value were noted. Statistical analyses of the discrete gait features and the GPS were carried out in SPSS (IBM SPSS Statistics 26, Armonk, NY, USA).

First, the normality of the continuous gait curves was evaluated using the built-in function of SPM. Since normality was not confirmed, the non-parametric SPM version (SnPM) was used with 10,000 iterations. Subsequently, sagittal plane differences on a vector-level were performed for the three conditions (baseline, VB feedback and VR feedback) using the multivariate analysis of variance (MANOVA) with an alpha level set at 0.05, taking the vector component covariance into account [25]. Each vector consisted of four components, namely the combined motions of the pelvis, hip, knee and ankle in the sagittal plane. In case of significance, post hoc joint-level comparisons were conducted.

Next, Hotteling’s test was used to evaluate differences between the different conditions with an alpha level set at 0.0167 (i.e., 0.05/3, because of the Bonferroni correction for multiple testing in the comparison of the three conditions). In case of a significant difference between conditions, a post hoc *t*-test was performed with a Bonferroni adjustment. The alpha level for the post hoc *t*-test was 0.004 (i.e., 0.05/(3 × 4), because of the Bonferroni correction for multiple testing in the comparison of the three conditions, for each of the four continuous waveforms, namely pelvis, hip, knee and ankle). The size of the significant cluster needed to be at least 3% to be considered clinically relevant [26].

Secondly, the data of the discrete gait features and gait deviation indices were checked for normal distribution using the Shapiro–Wilk test of normality. Since the collected data proved not to be normally distributed, the median and interquartile range (IQR) were reported, and a Friedman test was applied to test for differences between conditions. The alpha level was set at 0.007 (0.05/7, after Bonferroni correction for 7 gait features). Post hoc analyses with a Bonferroni correction for multiple comparison were performed with an alpha level of 0.002 (0.05/(7 × 3), after an additional Bonferroni correction for the three comparisons between conditions).

Finally, descriptive statistics (median, IQR and data frequency) were provided for the results of the clinical examination data.

## 3. Results

Thirty-one children with CP completed the study protocol on the treadmill. Two participants were excluded because they lacked sufficient cognitive skill to understand the feedback instructions as well as an adequate gait speed to be able to adjust their gait during training on a treadmill. As a result, 29 children were included in the current study.

### 3.1. Patient Demographics

The median age of the 29 enrolled children was 11.02 years (IQR 9.1–13.0). There were 20 male and nine female participants. Out of the 29 children, 21 were classified as GMFCS I (72.4%) and eight as GMFCS II (27.6%). One side was affected in 16 subjects (55.2%) and 13 (44.8%) children had bilateral CP. In the clinical examination, a hip extension deficit on the targeted side was found in three patients with a maximum of 10°. Seven patients were found to have a passive knee extension deficit with a maximum of 20°. Mean overground walking speed was 1.07 m/s (±0.20) and the mean walking speed on the treadmill was 0.58 m/s (±0.13).

Additional results on clinical examination can be found in Table 2.

### 3.2. Continuous Gait Curves

Results of the analysis of sagittal plane kinematics on a vector-level (i.e., combining all sagittal joint motions) are shown in Figure 3. A statistically significant improvement was observed both after VB and VR feedback (*p* < 0.001) when compared to the baseline gait curve, except during the midswing phase. The observed difference between VB and VR feedback did not exceed the predefined critical *p*-value (*p* = 0.02).

Results of the post hoc analysis are presented in Table 3. Continuous waveforms of pelvic and hip kinematics are shown in Figure 4. At the level of the pelvis, a statistically significant increase in anterior pelvic tilt was found at the end of loading response and during midstance, as well as during the pre- and initial swing phase in both feedback conditions compared to the baseline condition (*p* < 0.001). No significant difference was found for pelvic kinematics when comparing both feedback conditions. Hip extension did not change significantly after VB feedback. VR feedback, however, yielded a significantly improved hip extension during terminal stance when compared to baseline (*p* < 0.001). Yet, no significant change was seen between both conditions.

Results on knee and ankle kinematics and ankle kinematics are found in Table 3 and Figure 5. VB feedback showed a significant decrease in knee flexion during the initial swing phase when compared to the baseline (*p* = 0.001). After VR feedback, a decrease in knee flexion at terminal swing and initial contact (*p* = 0.002) and during mid and terminal stance (*p* < 0.001) was seen. No significant difference was found between both feedback conditions. Furthermore, at the ankle level, a reduced dorsiflexion at initial contact, loading response, midstance and pre-, initial and mid-swing phase was found after VB feedback (*p* < 0.001). The same results were observed after VR feedback, except that there was no change during loading response with VR feedback. Comparison of both feedback conditions yielded no significant difference.

The results based on the kinetic gait curves of the hip moment are presented in Figure 6. One patient was excluded from this analysis, due to lack of valid kinetic data. Assessment of the results showed a statistically significant increase in internal extension moment during loading response and a significant decrease in internal extension moment during terminal swing when VB was given compared to the baseline measurements (*p* = 0.0001 and *p* = 0.0008, respectively). After VR feedback, a statistically significant increase in internal extension moment during loading response and a significant decrease in flexion moment during initial swing were seen (*p* = 0.0007 and *p* = 0.0001, respectively). More importantly, a significant increase in the internal flexion moment was also found during terminal stance when VR feedback was given (*p* = 0.0017). No significant differences were found between both feedback conditions.

### 3.3. Discrete Gait Parameters

Results of discrete gait parameters are presented in Figure 7. At baseline, the median minimal hip angle was 6.15° (IQR −0.61–12.11). This angle decreased to 1.35° (IQR −4.34–11.28) after VB feedback and to −0.40° (IQR −3.95–9.03) after VR feedback. The improvement was found to be statistically significant both after VB and after VR feedback when compared to baseline. There was no statistically significant difference between VB and VR feedback conditions.

Furthermore, the timing within the gait cycle at which transition of internal hip extension moment to hip flexion moment occurred (hip moment 0), was defined and expressed as a percentage of the gait cycle. The same patient that was excluded from the kinetic gait curves analysis, was also excluded from this analysis due to the same reason. The median hip moment 0 was 40.32% (IQR 37.46–42.12) at baseline. After VB feedback, this percentage was 38.42% (IQR 33.94–42.99) and after VR feedback 37.05% (IQR 31.76–41.16). This change was not found to be statistically significant (*p* = 0.0595).

### 3.4. Gait Profile Score

The median GPS at baseline was 9.62° (IQR 8.70–10.90). There was an increase in GPS both after VB (10.24°) as well as after VR feedback (10.07°). The difference was statistically significant for VB feedback when compared to baseline, but not for VR feedback.

Analysis of the pelvic sagittal GVS showed a statistically significant increase after both VB and VR feedback from a median value of 6.62° (IQR 4.55–11.37) at baseline to 10.81° (IQR 5.58–16.27) and 10.20° (IQR 7.17–14.66), respectively. No significant difference between both feedback conditions was found. Assessment of the sagittal GVS of the hip and knee showed no significant differences after either feedback condition when compared to baseline. The median sagittal GVS of the ankle was 5.41° (IQR 4.35–7.99) at baseline with a significant increase after both VB (6.56° with IQR 5.50–9.61) as well as VR (7.20° with IQR 5.88–10.28). No significant difference was found between both conditions. Results are shown in Figure 6.

## 4. Discussion

This study investigated the immediate effects of VB and VR feedback on gait in children with CP during treadmill training. The results revealed several positive effects of both VB and VR feedback on gait in children with CP. First, we observed an improvement of the targeted feedback parameter, i.e., a statistically significant increase in maximal hip extension after VB as well as VR feedback and an increased internal hip flexion moment during terminal stance after VR feedback. Although no direct object of feedback, beneficial effects on both knee extension and ankle plantar flexion were seen as well. More specifically, knee extension increased significantly at initial contact as well as at the mid- and terminal-stance phases after VB feedback and ankle plantarflexion improved during toe-off after both types of feedback. Noorkoiv et al. analyzed gait efficiency and lower body kinetics and kinematics in 58 adolescents with CP and found that reduced knee and hip extension during walking was significantly related to a less efficient gait [18]. In this line, our results suggest that a more efficient gait may be achieved with targeted feedback on hip extension during treadmill training. This is further supported by our finding of an earlier transition from internal hip extension moment to internal hip flexion moment during stance after VB and VR feedback, although it was not statistically significant.

Several previous studies have investigated the effects of different types of feedback on gait in children with CP. Improvements in walking speed, step and stride length were described, as well as increased ankle peak power, increased ankle range of motion and reduced knee flexion at initial contact with both visual as well as auditory or proprioceptive feedback [6,12,13,27]. More specifically, two studies examined the immediate effects of feedback on gait in children with CP during treadmill training. Booth et al. investigated the effect of VR feedback on gait in 25 children with CP. They found statistically significant improvements on step length, knee extension during late swing and ankle power generation when feedback was targeted on these specific parameters. Furthermore, increased knee extension during stance phase was seen, although children received no direct feedback on this parameter. Additionally, the largest improvement on step length was found when feedback was given on knee extension and not on step length. Feedback on kinematic and spatiotemporal parameters also influenced kinetic parameters with an increased ankle power generation during feedback on knee extension [14]. Van Gelder et al. gave VR feedback on hip and knee extension in 16 children with CP and found significant improvement of the targeted parameter in both conditions [15]. These results, together with our results, confirm the adaptability of gait in children with CP and the positive immediate effects of feedback during treadmill training. Moreover, it highlights the complexity of gait in children with CP as feedback directed at one parameter also influences other gait parameters.

Although the targeted feedback parameter significantly improved, we observed a deterioration of the overall gait performance as measured by the GPS when comparing feedback conditions to baseline values. This finding was statistically significant after VB feedback (9.62° to 10.24°) but not after VR feedback (9.62° to 10.07°). However, the change did not reach the minimally clinically important difference of 1.6°, as stated in the literature [28]. The lack of improvement of the GPS is most likely due to compensatory strategies that negatively affect the gait cycle, such as the increased pelvic anterior tilt during the majority of the gait cycle. It is to be expected that other compensations occur in both the coronal and transversal plane as well as at the level of trunk kinematics. These findings are supported by previous studies on the effects of feedback on gait in children with CP with no change of GPS after feedback due to compensatory strategies, such as increased hip abduction during swing, increased step width and increased trunk and pelvis ranges of motion [14,15]. When trying to reach a targeted goal, it is likely that children focus on that specific goal for which they are being rewarded, while other aspects of the gait are neglected. Further research is necessary to investigate whether these compensatory strategies are transferred to overground gait. Moreover, it would be interesting to explore how the VR tool can be adjusted to take into account and eliminate compensatory strategies.

When analyzing the differences between VB and VR feedback, we found a tendency to a slightly larger positive effect of VR feedback on maximal hip extension when compared to VB feedback. The observed differences between both feedback conditions were, however, small and not statistically significant. Nonetheless, a statistically significant improvement of the sagittal kinematics and kinetics of the hip during terminal stance was only observed after VR feedback. Although not directly targeted, VR feedback also yielded an improvement in sagittal knee kinematics with less knee flexion during stance phase. It is known that VR feedback has several advantages over VB feedback, although it is more demanding in terms of infrastructure and financial costs. Different aspects of VR are proven to influence children’s motivation in a positive way, through the more variable and unpredictable environment. VR challenges children to perform slightly above their usual degree of skill, stimulates competition and gives rewards when reaching a targeted goal [29]. Furthermore, VR feedback allows gait training at a higher intensity without the need for additional human resources. Additionally, previous studies suggest that a VR environment stimulates neuroplasticity by allowing multiple repetitions of gait-specific exercises and by activation of motion-related representation sites in the patient’s brain through virtual motion [30]. Golomb et al. confirmed these findings with functional MRI in three adolescents with spastic hemiplegia who received virtual training on the right hand. They found increased spatial extent of activated areas in motor-related regions postintervention, suggesting neuroplasticity [31]. Overall, we can conclude that both feedback conditions can be effective to improve a targeted gait parameter of children with CP during treadmill training. This study indicated a slightly, not statistically significant, larger effect of VR compared to VB feedback, which warrants that further investigation be performed on larger samples. At this stage, we would carefully suggest VR feedback as a preferred feedback strategy due to its slight benefits over VB feedback regarding improvement of the targeted feedback parameter as well as its other benefits, such as its repetitive nature and enjoyable and motivating environment. During VR feedback sessions, VB feedback can be used as additional feedback strategy to correct compensatory gait patterns caused by the primary VR feedback condition. Yet, the effects of such hybrid approach (combined VR and VB feedback) on gait need to be further investigated.

This study has several limitations. First, we only studied the immediate effects of VB and VR feedback on gait in children with CP. Therefore, no conclusions can be made on whether the effects persist after cessation of feedback or on the long-term effects of this type of gait training. We suggest that future research should include treadmill training programs of several weeks with targeted feedback and assessment before, during and after the program to assess possible long-lasting effects. Secondly, our study population of 29 children was relatively small, although exceeding the minimal number based on the power analysis. All included participants were functional children (GMFCS I or II) without severe hip flexion contractures, no to little spasticity in the psoas and rectus femoris and with sufficient cognitive abilities to understand and respond to the different feedback types. Our results can therefore not be extrapolated to the general CP population but must be interpreted in the correct context of functional CP children. It is to be expected that feedback on hip extension in children with severe spasticity in the hip flexors or a large hip extension deficit respond less well to this type of feedback than the current study population. Research on a larger and more heterogenic study population is of interest to further determine the effects of different feedback strategies on gait in children with CP. Another limitation of our study is the fact that verbal feedback commands were not strictly standardized. Various statements of VB feedback can influence the gait differently. Furthermore, our research showed coinciding positive effects of improved hip extension with improved ankle plantar flexion at push-off with VB feedback. It would be interesting for future research to see which commands, either on the hip or on the ankle, produce the best effects on gait in children with CP and which ones are best understood. On the other hand, using VB feedback statements that are slightly adjusted according to the children’s specific needs acknowledges child-specific problems and capabilities for this heterogenous patient group, which is in our opinion crucial. Bias of potential learning and fatigue effects was minimized by ensuring equal distribution of selected gait cycles for further analysis throughout each intervention session. However, further research on this topic would most certainly be of interest to be able to take this confounding factor better into account. Finally, we chose to focus only on sagittal gait parameters. This means no data on spatiotemporal parameters nor coronal or axial gait kinematics and kinetics were assessed, except for the kinematic data needed to calculate the GPS. Therefore, we assume no improvement of the GPS occurred due to compensatory strategies in the coronal and axial planes. However, no definite conclusions can be made.

## 5. Conclusions

This study confirms the positive effects of both VB and VR feedback on a targeted gait parameter in children with CP during treadmill training. It also highlights the complexity of gait with both positive side effects on gait parameters other than the targeted parameter, as well as negative compensatory strategies. These effects contribute to a small deterioration of the overall gait performance, as measured by the GPS. Furthermore, the results of this study showed that both feedback conditions are effective with a slight advantage of VR over VB feedback. Together with the additional advantages of VR being more challenging and rewarding, in our experience, we suggest VR feedback as the main feedback strategy with VB feedback as additional feedback strategy to correct compensatory gait patterns caused by the primary feedback condition.

## Figures and Tables

**Figure 1 children-11-00524-f001:**
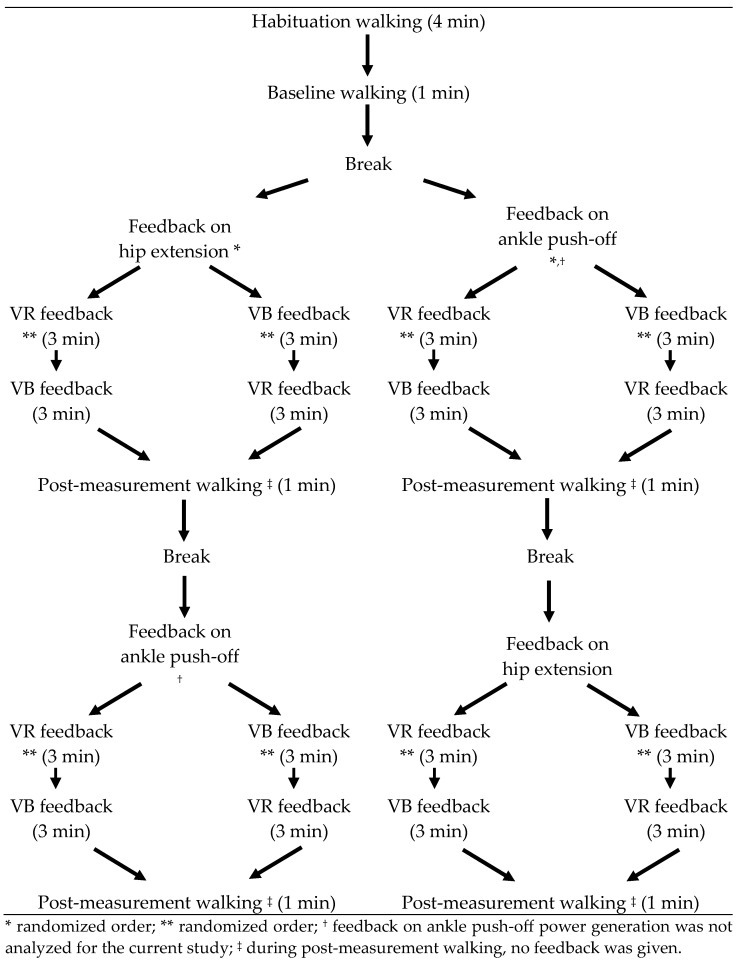
Study Protocol.

**Figure 2 children-11-00524-f002:**
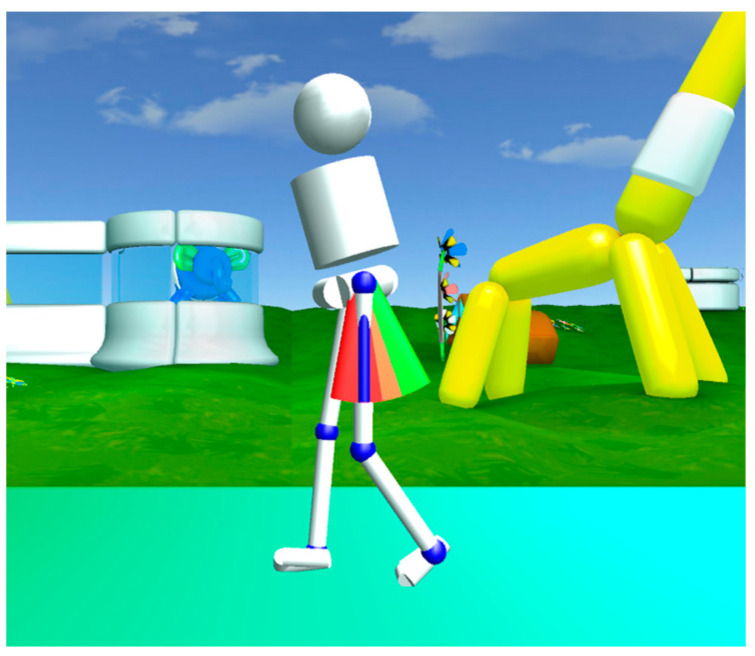
Representation of VR feedback on hip extension in which the child’s movements are represented by an avatar in a stimulating environment. The green bars signify the targeted goal for hip extension. When reaching the green bars, the participants were rewarded with points.

**Figure 3 children-11-00524-f003:**
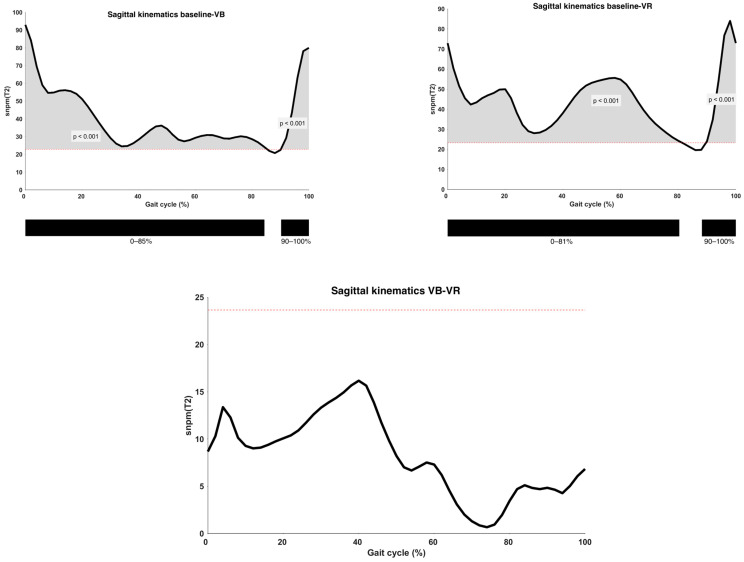
Analysis of sagittal kinematics between different types of feedback conditions on the vector level, with indication of significant clusters. α = 0.0167; VB = Verbal feedback; VR = Virtual Reality feedback; SnPM(T2) = Statistical Non-Parametric Mapping Analysis (Hotteling’s *t*^2^-test); Black bar below graphs = cluster within the gait cycle with significant difference between both feedback conditions, based on SPM analysis and exceeding more than 3%; Red dotted line = cut-off above which the difference is defined as significant.

**Figure 4 children-11-00524-f004:**
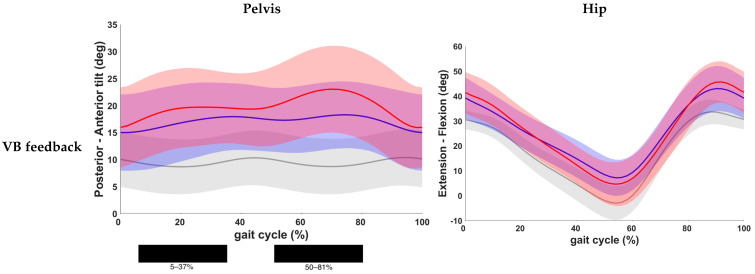
Comparison of kinematic waveforms of the pelvis and hip during the different feedback conditions, with indication of significant clusters in the gait cycle. Red = VB feedback; green = VR feedback; blue = baseline; grey = normative data of typically developing children; light red, green and grey zones = ±1 standard deviation; black bar below graphs = cluster within the gait cycle with significant difference between both feedback conditions, based on SPM analysis and exceeding more than 3%. Definition of Y-axis is indicated in the graph as (negative)–(positive) values.

**Figure 5 children-11-00524-f005:**
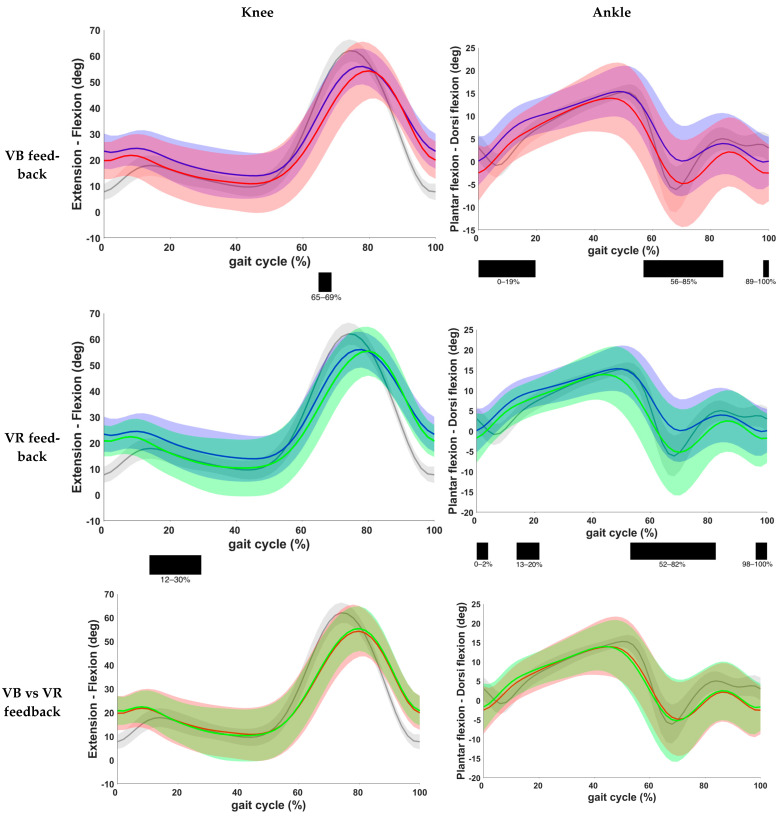
Comparison of kinematic waveforms of the knee and ankle during the different feedback conditions, with indication of significant clusters in the gait cycle. Red = VB feedback; green = VR feedback; blue = baseline; grey = normative data of typically developing children; light red, green and grey zones = ±1 standard deviation; black bar below graphs = cluster within the gait cycle with significant difference between both feedback conditions, based on SPM analysis and exceeding more than 3%. Definition of Y-axis is indicated in the graph as (negative)–(positive) values.

**Figure 6 children-11-00524-f006:**
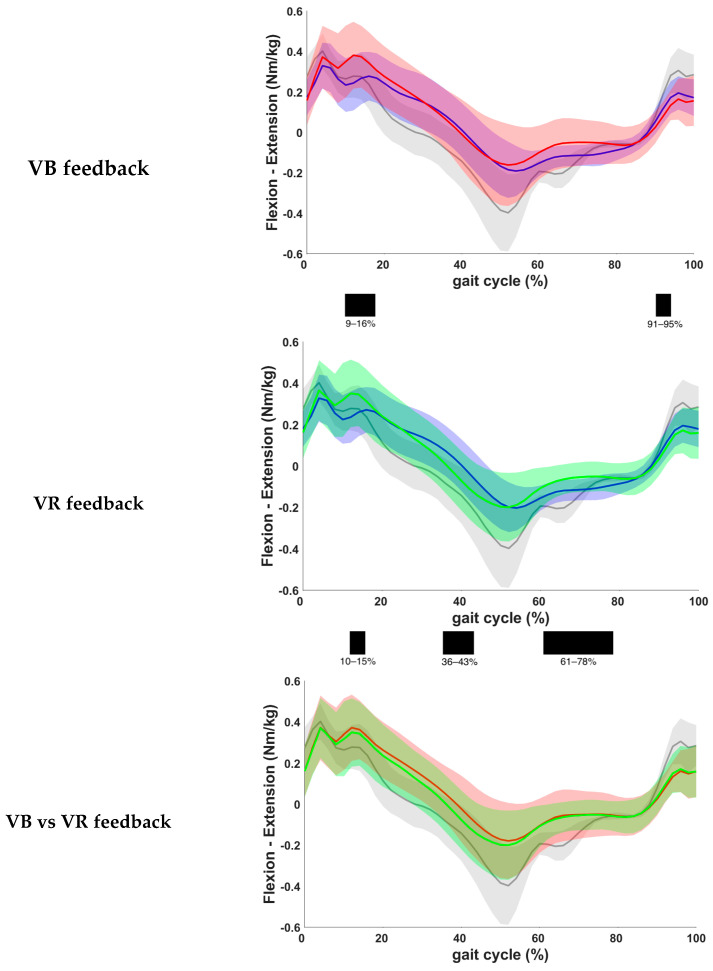
Comparison of kinetic waveforms of the internal hip moment during the different feedback conditions, with indication of significant clusters in the gait cycle. Red = VB feedback; green = VR feedback; blue = baseline; grey = normative data of typically developing children; light red, green and grey zones = ±1 standard deviation; black bar below graphs = cluster within the gait cycle with significant difference between both feedback conditions, based on SPM analysis and exceeding more than 3%. Definition of Y-axis is indicated in the graph as (negative)–(positive) values. The increased hip moments at loading response that are observed in children with CP compared to typically developing children can be attributed to a less-controlled placement of the foot, resulting in sudden altered directions of the ground reaction forces.

**Figure 7 children-11-00524-f007:**
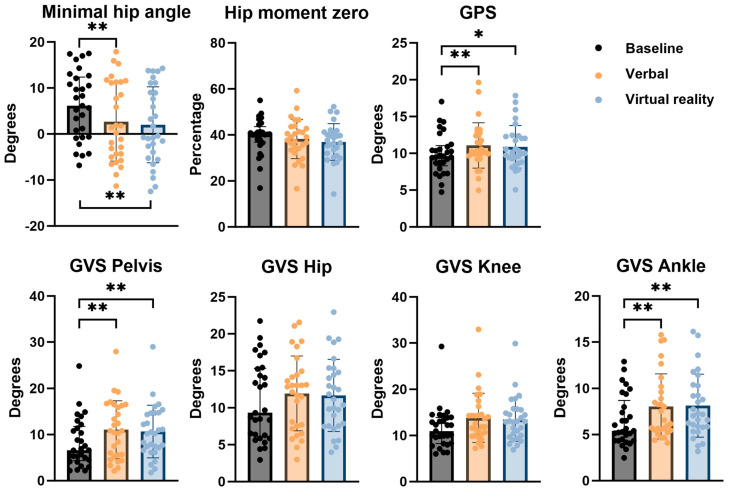
Comparison of results on discrete gait parameters for the different types of feedback conditions, with indication of significant differences. * 0.002 < *p* < 0.05; ** *p* < 0.002; GVS = Gait Variable Score; GPS = Gait Profile Score.

**Table 1 children-11-00524-t001:** Schematic overview of data processing.

Initial Data Set ^1^		
Baseline (1 min)	VB feedback (3 min)	VR feedback (3 min)
10 steps	30 steps	30 steps
↓		
1st quality check exclusion of obvious outliers *		
↓		
2nd quality check further exclusion of steps that deviated most from the averaged picture *^,^**		
↓		
Final data set ^2^		
Baseline (1 min)	VB feedback (3 min)	VR feedback (3 min)
7 steps	21 steps	21 steps

^1^ Ten steps per minute to ensure equal distribution of selected steps; ^2^ Seven steps per minute to ensure equal distribution of selected steps; * Through visual inspection; ** If >7 steps/min remain; VB = Verbal feedback; VR = Virtual Reality feedback.

**Table 2 children-11-00524-t002:** Results of clinical examination.

Parameter (*n* = 29)		Median	IQR
Passive ROM (°)	Hip extension	0	0–0
	Knee extension	0	0–0
	Popliteal angle, bilateral	−65	−70–−55
	Popliteal angle, unilateral	−65	−70–−55
	ADKE	10	5–10
	ADKF	15	10–20
MAS	Hip flexors	0	0–0
	Hamstrings	1.5	1–1.5
	Rectus femoris	0	0–1
	Soleus	1.5	1–1.5
	Gastrocnemius	1.5	1.5–2
Strength (MRC)	Hip extensors	4	4–5
	Hip flexors	4	4–5
	Hip abductors	4	3–4
	Hip adductors	4	4–5
	Knee extensors	4	4–5
	Knee flexors	4	3–4
	Gastrocnemius	3	3–4
	Gastrocnemius + soleus	3	3–4

Only results from the targeted leg are shown; IQR = Interquartile Range; ROM = Range Of Motion; ADKE = Ankle Dorsiflexion with Knee Extension; ADKF = Ankle Dorsiflexion with Knee Flexion; MAS = Modified Ashworth Scale.

**Table 3 children-11-00524-t003:** Post hoc analysis of the kinematic waveforms in the sagittal plane.

		Baseline—VB	Baseline—VR	VB—VR
Pelvis	n-clusters	2	2	-
	(*p*-value)	(*p* < 0.001; *p* < 0.001)	(*p* < 0.001; *p* < 0.001)	
	% cluster	5–37%; 50–81%	6–24%; 52–82%	-
Hip	n-clusters	0	1	-
	(*p*-value)		(*p* < 0.001)	
	% cluster	-	31–50%	-
Knee	n-clusters	1	1	-
	(*p*-value)	(*p* = 0.001)	(*p* < 0.001)	
	% cluster	65–69%	12–30%	-
Ankle	n-clusters	2	3	-
	(*p*-value)	(*p* < 0.001; *p* < 0.001)	(*p* = 0.002; *p* = 0.001; *p* < 0.001)	
	% cluster	0–19%; 56–85%	0–2% *; 13–20%; 52–82%; 98–100% *	

VB = verbal; VR = virtual reality; n-clusters = number of significant clusters; % cluster = extent of the cluster, expressed as a percentage of the gait cycle; * considered as one cluster.

## Data Availability

The data presented in this study are openly available in the Research Data Repository (RDR) of the Catholic University of Leuven at https://rdr.kuleuven.be/dataset.xhtml?persistentId=doi:10.48804/JUNNWW.

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
