# Peer review of "A Comparison of the Immediate Effects of Verbal and Virtual Reality Feedback on Gait in Children with Cerebral Palsy"

_children, 2024, doi:10.3390/children11050524_

Round 1
Reviewer 1 Report
Comments and Suggestions for Authors
The immediate effects of verbal and virtual reality feedback on sagittal plane kinematics and kinetics in children with CP are analysed in this paper.
The reasearch question is clearly presented, the methods are well described, and the discussion has a clear structure.
The figures 3, 4 and 5 are difficult to understand. I assume that the grey line are Norm data. And I assume that the graphs are +/-1 standard deviation. These details are missing in the captions of the figures. Indication of positiv (anterior tilt, hip flexion) movements is as well missing.
The nomenclature of the joint moments should be specified. I assume that the authors presented the data of the internal hip flexion moment. Unfortunately, that is not formulated precisely in the text (e.g. p.6 line 193, p.12, line 276, figure 5).
On page 3 line 113 seems to be a mistake in the software name and a citation missing
Comments on the Quality of English Language
The paper is well written.
On page 5 line 176 it should be: All data "were" imported ..
Author Response
Thank you for reviewing our manuscript and for your response. We appreciate the time and effort it has taken to thoroughly read the article and provide us with feedback.
We understand Figures 3-4-5 were not thoroughly enough explained. As such, extra information is added in the reference under each of the figures as to make sure readers are able to fully comprehend the graphs.
We also adjusted 'hip flexion moment' to 'internal hip flexion moment' where this was missing, as we comprehend that leaving out 'internal' is confusing.
Our apologies for the missing reference in the software name on page 3, it is now added after the sentence.
Reviewer 2 Report
Comments and Suggestions for Authors
First of all, thank you for allowing me to review the manuscript entitled: “A Comparison of The Immediate Effects of Verbal and Virtual Reality Feedback on Gait In Children With Cerebral Palsy”
This paper is a single-centre study on gait training in children with cerebral palsy using verbal feedback and virtual reality. The study investigated the effects of both types of feedback on specific gait parameters, such as sagittal hip kinematics and hip kinetics.
After reading in depth the manuscript, I would like to make some comments and ask the authors several questions about.
I think that dividing the abstract into sections could improve its clarity and organisation (background, material and methods, results, conclusions).
Introduction
- line 33 correct the end of the sentence so as not to break the syllable, please review the whole document.
- In line 38 ,references 2 and 3 should go together in the same parenthesis. Please correct. Review the whole document, lines 47, 53... references 6, 8, 10. 12...
-In my view the authors should clarify and highlight from the outset the main objective of t
Material and methods
- Although it is mentioned that the study was conducted in accordance with the Declaration of Helsinki and received the approval of the Medical Ethics Committee of the University Hospital Leuven, it would be useful to provide more details on the ethical procedures followed during recruitment and participation of subjects, as well as any specific considerations related to informed consent and privacy of participants.
Discussion
-The discussion analyses in detail the results obtained in the study, highlighting both the positive effects and the limitations observed.
Please delete: Appendix A
The appendix is an optional section that can contain details and data supplemental 452
to the main text—for example, explanations of experimental details that would disrupt 453
the flow of the main text but nonetheless remain crucial to understanding and reproduc- 454
…. and
Appendix B
All appendix sections must be cited in the main text. In the appendices, Figures, Ta- 459
bles, etc. should be labeled starting with “A”—e.g., Figure A1, Figure A2, etc.
References
- the reference section must be corrected, the reference number cannot appear twice.
Author Response
Thank you for taking the time to thoroughly review our manuscript. We appreciate your feedback and have revised our manuscript accordingly, as we feel that your suggestions indeed improve the article.
As for the abstract, we understand that dividing the abstract into sections would make it more comprehensive for readers. However, the abstract was written according to the instructions of the journal and as such, we feel we cannot adjust the abstract to meet your suggestion.
We appreciate your attention to the overall lay-out of the paper and have adjusted the format so that no words are broken at the end of a sentence and that multiple references are placed together in the same parenthesis.
As to your last feedback on the introduction: unfortunately, your suggestion doesn't seem fully finished and as such, it was difficult for us to understand the exact meaning. However, we have revised the first part of the introduction as we understand that the outset of the article wasn't clear from the first few sentences. With this, we hope we have met your feedback sufficiently.
We have added extra information to the section on inclusion of participants and the approval of the Medical Ethical Committee to further clarify the procedures taken to ensure privacy of the participants.
Finally, we would like to thank you to review the full bibliography and to also provide feedback on this section. We have thoroughly analysed all the references and can confirm that no reference number appears twice.